# In Vitro Model of Human Trophoblast in Early Placentation

**DOI:** 10.3390/biomedicines10040904

**Published:** 2022-04-15

**Authors:** Darina Bačenková, Marianna Trebuňová, Daša Čížková, Radovan Hudák, Erik Dosedla, Alena Findrik-Balogová, Jozef Živčák

**Affiliations:** 1Department of Biomedical Engineering and Measurement, Faculty of Mechanical Engineering, Technical University of Košice, 04200 Košice, Slovakia; marianna.trebunova@tuke.sk (M.T.); radovan.hudak@tuke.sk (R.H.); alena.findrik.balogova@tuke.sk (A.F.-B.); jozef.zivcak@tuke.sk (J.Ž.); 2Centre for Experimental and Clinical Regenerative Medicine, The University of Veterinary Medicine and Pharmacy, 04181 Košice, Slovakia; dasa.cizkova@uvlf.sk; 3Department of Gynecology and Obstetrics, Faculty of Medicine, Pavol Jozef Šafarik Univerzity Hospital AGEL Košice-Šaca, Pavol Jozef Šafarik University in Košice, 04015 Košice-Šaca, Slovakia; erik.dosedla@nke.agel.sk

**Keywords:** trophoblast stem cells, trophoblast invasion, organoids

## Abstract

The complex process of placental implantation and development affects trophoblast progenitors and uterine cells through the regulation of transcription factors, cytokines, adhesion receptors and their ligands. Differentiation of trophoblast precursors in the trophectoderm of early ontogenesis, caused by the transcription factors, such as CDX2, TEAD4, Eomes and GATA3, leads to the formation of cytotrophoblast and syncytiotrophoblast populations. The molecular mechanisms involved in placental formation inside the human body along with the specification and differentiation of trophoblast cell lines are, mostly due to the lack of suitable cell models, not sufficiently elucidated. This review is an evaluation of current technologies, which are used to study the behavior of human trophoblasts and other placental cells, as well as their ability to represent physiological conditions both in vivo and in vitro. An in vitro 3D model with a characteristic phenotype is of great benefit for the study of placental physiology. At the same time, it provides great support for future modeling of placental disease.

## 1. Introduction

Pathology in the process of deficient placentation is associated with the development of serious disorders during pregnancy. Failure to form a deep placental bed, placental–uterine interface and fetal growth restriction may result in a potential miscarriage.

The defective process in the trophoblast cytotrophoblast and differentiation invasion are associated with preeclampsia. The etiology of a large number of diseases inside the female reproductive tract is not fully understood due to the fact that it is relatively difficult to monitor them in ex vivo conditions. The chapters present the gradual development of placenta from trophoblast invasion and early hypoxic states to detailed differentiation of individual trophoblast cell subtypes, the transcription factors involved in these processes and immune processes. In the second part of the article, an in vitro model of the placenta and its cell populations are chronologically described. The review is beneficial to its readers, as it provides detailed description of the multifactor-controlled placentation process and its subsequent steps. Current knowledge in this field is evolving, and the topic has already been addressed by several authors [1,2,3]. Established in vitro models of trophoblastic invasion were reviewed by authors Abbas et al. The authors also focused on comparing traditional and dynamic models through microfluidic tests [2]. Currently, the technologies of controlled hydrogel matrices in organoid culture are available. Different types of organoids intended for tissue recapitulation in the female reproductive system were derived, including the ovaries, cervix, fallopian tubes and endometrium using a Matrigel matrix [3].

The implantation process and the gradual placenta development affect trophoblast progenitors and uterine cells through the regulation of transcription factors (TFs), cytokines, adhesion receptors and their ligands [4]. Many researchers utilize cancer cell lines, primary cells, tissue explants and three-dimensional (3D) organoid technology in order to study human implantation via an available in vitro model. New information has been gained based on a study of human trophoblast differentiated cells, which are involved in early implantation and placental formation and are monitored both in vivo and in vitro. The acquired knowledge will then be applied in the optimization of 3D culture forms and in the use of stem cells in hydrogel matrices, such as Matrigel. The most characteristic cell types suitable for organoids are human pluripotent embryonic stem cells (ESCs), human-induced pluripotent stem cells (PSCs) and organ-restricted adult stem cells (ASCs) [5,6] (Table 1). This innovative 3D culture technology provides a benefit in recognizing stem cell properties and is therefore suitable for in vitro modeling of organs and pathological processes “in a bowl” [5,7]. Significant progress has been made in designing various protocols for human organoids, such as the brain [8,9], lungs [10], liver [11], kidneys [12,13], prostate [14], retina [15] and gastrointestinal tissues [16,17]. In contrast to the precise culture conditions of the above-mentioned organoids, organoids simulating the human placenta have begun to attract more attention. It is very difficult to study such unique environment of the fetus in humans [18]. The in vitro model of the placenta will benefit from understanding the etiology of female reproductive tract disorders. However, it is difficult to study the aforementioned disorders, and thus, they are not well known. 

## 2. Human Placenta Development

The placenta is a transitional organ with several functions, which mainly exchange nutrients and gases between the mother and the fetus. It further acts as a specific endocrine organ that generally supports fetal development. The placenta produces a number of hormones, such as estrogen, progesterone and human chorionic gonadotropin (hCG). Many physiological parameters change during a woman’s pregnancy. The pituitary gland is enlarged during pregnancy in order to produce oxytocin. Oxytocin level increases during pregnancy. Serum prolactin levels increase significantly in the first trimester and are affected by increasing serum estradiol concentrations [19]. Extraembryonic tissues from the placenta develop dynamically during pregnancy [3,20]. In order to sustain the pregnancy, it is necessary to form spiral arteries in the decidua and thus assure an optimal supply of nutrition and oxygen [21].

### 2.1. Trophoblast Invasion

Trophectoderm (TE), the outer layer of the human blastocyst, is a source of trophoblast progenitor cells. Trophoblast invasion of the uterus is the first step to implantation of the human blastocyst. The process is also necessary for transforming the uterine spiral arteries [22]. Trophoblast stem cells (TSCs) and trophoblast progenitor cells (TPCs) differentiate into more specialized trophoblast populations at an early stage of development. TSCs are further classified into two types of cells—mononuclear cytotrophoblasts (CTBs) and multinucleated primitive syncytium (PS) [20,23]. CTBs have the ability to proliferate, differentiate and fuse on multinuclear syncytiotrophoblasts (STBs), thereby promoting syncytial growth during ontogenesis [24]. Early development and accurate differentiation of CTBs toward the different subtypes of extravillous trophoblast (EVTs) is a prerequisite for further physiological pregnancies. Invasive EVTs are made up of different types of specialized cells that have the potential to differentiate and proliferate [25,26]. CTBs form a multinucleated STBs through cell fusion. Therefore, cytotrophoblasts form the basis of the STBs and reside on the basement membrane that separates them from the villous stroma [27]. The proliferative activity of these trophoblast subtypes varies considerably. While STBs have a very limited proliferation, CTBs have a high proportion of proliferating cells [28] (Figure 1).

### 2.2. Early Hypoxic Placental Environment 

It is very interesting to point out that the hypoxic environment exists in the early placental period. Current oxygen levels play an important role in managing the differentiation process, which leads to the invasion of cytotrophoblast into the uterus. In the early stages of placental development, the environment is relatively hypoxic and affects more cytotrophoblast proliferation than invasive differentiation along the invasive pathway [29]. Cells respond to change in hypoxic conditions through transcription factors by hypoxia-inducible factors (HIF). HIF transcription factors affect placentation and vascularization by activating the vascular endothelial growth factor (VEGF) gene expression. The HIF gene family includes three alpha subunits, HIF1A, HIF2A and HIF3A, and three beta subunits, HIF1B, HIF2B and HIF3B. The HIF1A, HIF2A subunits are affected by oxygen, and the beta subunit of HIF1B is constitutively expressed. Following hypoxia, the HIF alpha subunits accumulate in the nucleus and dimerize with HIF1B, allowing them to bind to the DNA and initiate transcription of their target genes. The above-described process leads to targeted gene activation and response to hypoxia, including pathways in which the cellular need for O2 is reduced, thus representing adaptation and the resulting reoxygenation. HIFs affect the implantation and placentation processes in particular, as well as increasing vascular permeability and angiogenesis [30]. CTBs interfere with the uterine wall and plug the maternal vessels, helping to maintain a state of physiological hypoxia [31]. Maternal spiral arteries that supply the placental site are blocked by CTB plugs [4]. In early placental development, the intrauterine oxygen tension of maternal decidua is 3%, and the oxygen tension of the myometrium is about 10%. This standing oxygen gradient most likely supports and directs the invasion of EVT into the decidua and myometrium, where it remodels the spiral arteries of the uterus [32]. The placenta develops improved mechanisms to protect against oxidative damage [4]. Between 10 to 12 weeks’ worth of human pregnancy blood starts to flow to the intervillous space. While high oxygen gradient promotes EVT invasion, low oxygen causes hypoxia-inducible factor 1a (HIF-1a)-dependent EVTs proliferation leading to the rapid growth of the placenta in the early stages of pregnancy [33]. The oxygen gradient values in the early period of placentation are also interesting for the in vitro placental and trophoblast model. 

### 2.3. Cytotrophoblast Differentiation 

In the early period, vacuoles appear on the PS. Upon fusion, vacuoles form a network of lacunar spaces that eventually breach the maternal uterine capillaries and create discontinuous maternal blood sinusoids. The next stage is the development and morphogenesis of placental villi. At the time of PS expansion, a series of proliferative CTBs outgrow the expanding syncytial mass and form primary villi (PV). The PV grows toward the underlying maternal decidua. The process rapidly expands by the continuous proliferation and cell fusion of the developing villous cytotrophoblasts (vCTBs) [20,29,33]. Stem cells in the vCTBs differentiate into proliferative proximal cell column trophoblasts (pCCTs), which bind to the villi of the maternal decidua and are proliferatively active [34]. After anchoring the villi, pCCTs further develop into distal cell column trophoblasts (dCCTs). These dCCTs cells have a function in the process of anchoring the developing embryos in the uterine wall [35]. dCCTs in EVTs migrate deeply into the decidua basalis and the first third of the underlying myometrium and into maternal spiral arteries where they adopt a vascular phenotype, and EVT cells remodel the maternal spiral [2,36,37]. The process of placentation is characterized by a deep invasion of trophoblasts, which is necessary to ensure increased blood flow during a longer intrauterine period [36]. Differentiation of EVT subtype progenitors residing on the basement membrane of CCTs leads to interstitial cytotrophoblasts (iCTB) and endovascular cytotrophoblasts (eCTB) relocated to maternal helical arteries. iCTBs in the decidual stroma colonize blood vessels from the outside and communicate with uterine cells of various types, such as decimal stromal cell macrophages and uterine natural killer (NK) cells [29,34].

The eCTB differentiated subset of EVTs form spiral uterine arteries with vascular transformation. Cells further interact with stromal cells, glands, arteries, macrophages and decidual (dNK) [2,25]. The decidualization process and its pre-decidual change are observed in the secretory phase of the human menstrual cycle. As iCTV progresses, the cells migrate to the lumen of the spiral arterioles and replace the endothelial lining of the uterus. iCTV are found in the smooth muscle walls of these vessels. During pregnancy, in which placental cells remodel the uterine arterioles, the decidual and inner thirds of the myometrial parts of these vessels are included [38]. Subsequently, the diameter of the arterioles expands, thereby increasing the blood flow needed to support the rapid growth of the fetus later in the pregnancy. It can be assumed that unsuccessful endovascular invasion may, in some cases, lead to miscarriage, whereas insufficient vascular perfusion to the appropriate depth is associated with preeclampsia and a subset of pregnancies in which fetal growth is restricted [31]. The placenta is an evolutionarily diverse organ. Placentation in mammals is highly variable, especially in the extent of invasion into the uterus by trophoblast cells [39]. In humans, the ratio of placenta to neonatal weight is 6:1, which indicates the great importance of placental function. The animal model of a mouse can be used to study the basic properties of the placenta. The mouse is an ideal model for studying the placenta with similar properties to the temporary organ because, like simian primates and humans, mice have a hemochoric type of placenta. This type of placenta is characterized by direct contact between the trophoblast and maternal blood without separation of the endothelium or epithelium [2,36,40] (Figure 2).

### 2.4. Transcription Factors, Cytokines in Early Placentation

Pregnancy is characterized by elevated levels of specific cytokines at the fetal–maternal interface. The complicated differentiation process of the trophoblast precursors in the TE of early ontogenesis is modulated by TFs, which act as transcriptional promoters, leading to the formation of the fetal part of the placenta. TFs, such as caudal-type homeobox 2 (CDX2), transcriptional enhancer factor 3 (TEAD4), Eomesodermin/T-box brain protein 2 (Eomes), GATA binding protein 3 (GATA3) and Transcription factor AP-2gamma (TFAP2C), are the most important factors in the process of TE specification in mice. The molecular mechanism of TE specification in humans is less exact. There is a similar molecular mechanism to the expression of some key TFs, including CDX2, TFAP-2C, transcription factor AP-2alpha (TFAP2A), chorion specific transcription factor (GCMa) and GATA3, which are found in the blastocyst-stage human TE progenitors [41]. The placenta-specific TFs are involved in further trophoblast development through the transcription of placenta-specific genes. At present, not all key TFs involved in the trophoblast self-renewal and differentiation have been sufficiently clarified. CDX2 plays a crucial role in the development of fetus and its perinatal tissues. Early preimplantation failure and thus embryonic lethality have been found to cause early CDX2 blockade. CDX2 is one of the most common feature genes that confer individual identity to specific segments through the possession of a “homeobox” DNA binding motif encoding 60 amino acids that acts as a transcriptional regulator of “downstream” genes [42]. CDX2 is important for TE development in preimplantation mouse embryos and human TSC-like stem cells. CDX2 expression is manifested in TSC at an early stage of placentation. CDX2 TF is less present in CTB progenitors. Selective expression of CDX2 in human CTB progenitors was observed in the first trimester [6,43]. The role of CDX2 and cadherins in the regulation of epithelial-mesenchymal transition (EMT) is currently hypothesized [44]. Trophoblast cells that result from vCTB differentiation proliferate the column trophoblasts with CDX2 negative and Neurogenic locus notch homolog protein 1 (NOTCH1) positive phenotypes. High expression of CDX2 ±, NOTCH1 + CTBs is present here [43]. Authors hypothesize that early TE progenitors leave the pluripotency state of undifferentiated cells relatively rapidly and adopt a phenotype comprising GATA2+, GATA3+, TFAP2A+ and TFAP2C+. A set of four genes as the trophectoderm four “TEtra” has been identified [41]. The GATA family of zinc-finger TFs regulates the critical steps in cell differentiation during vertebrate development [45]. GATA2 and GATA3 fulfill several properties of this claim and act conservatively in mammals, regulating the activation of trophoblast genes during placental development. Both GATA2 and GATA3 mRNAs are highly expressed in the human TE. GATA family members are selectively expressed within trophoblast progenitors during early mouse development and directly regulate the key genes. GATA2 and GATA3, as well as EVT, were confirmed to be highly expressed in cCTBs. 

At the beginning of pregnancy, there is a TEAP4-binding Yes-associated protein (YAP) coactivator, which plays a key role in the proliferation and expression of TSC villous trophoblastic epithelium progenitors. Thus, the Hippo signaling pathway could be very active in placental development. Other authors also point out that TEAD4 is critical for the self-renewal of human and mouse TSCs [46]. Therefore, YAP may not only trigger trophoblast proliferation by promoting the expression of cell cycle and stemness-associated genes, but also by inhibiting differentiation. In particular, the downregulation of gene characteristic of STBs and cell fusion regulators may be crucial to the expansion of YAP-dependent vCTB in early pregnancy. The effects of YAP on STB formation could be indirect. Therefore, YAP-TEAD4-EZH2 complexes may be required to inhibit the transcription of a subset of genes, STB-specific regulators, with the resulting effect of suppressing cell fusion. YAP–TEAD4 complexes bind to stemness and cell cycle genes, increase their expression and thereby promote vCTB/TSC expansion. Concurrently, YAP–TEAD4 complexes also inhibit cell fusion and STB marker expression by forming gene-repressive complexes in their promoter regions [47]. TFAP-2A is also involved in the process of EVT invasion and is identified by its expression in EVT. The TFAP-2A acts as a sequence-specific DNA-binding transcription factor, recognizing and binding to the specific DNA sequence and recruiting the transcription machinery. It is also characteristic of the invasive trophoblast cell lines. TFAP-2 alpha is involved in the cell-specific hormonal expression of syncytializing human trophoblasts. However, its role in invasive trophoblast differentiation remains largely unexplored [48]. Authors used immunofluorescence to examine first-trimester placental tissues and primary CTB pools. It was found that Neurogenic locus notch homolog protein 2 (Notch2) expression increased during EVT differentiation, which is characterized by the induction of marker genes, such as human leukocyte antigen G (HLA-G), Integrin alpha 1 (ITGA1) and ITGA5 [49] (Figure 3).

Extracellular signaling mechanisms regulate the individual and subsequent stages of trophoblast development. Molecules that are involved in the invasive processes of EVT are cell adhesion molecules (CAMs), such as integrins, cadherins and fibronectin, metalloproteinases (MMPs) and Tissue metalloprotease inhibitors (TIMPs) [50]. EVTs differentiation results in decreased E-cadherin expression in cell columns [35]. Cell adhesion is mediated through specific surface receptors known as integrins. The receptors, which may be heterodimeric αβ-type transmembrane glycoproteins, have a specific bond according to the type of α/β combination. CTBs express integrins to adhere to the extracellular matrix (ECM) components and modulate their integrin repertoire when they invade maternal tissues [51,52]. Proliferative CTBs form column CTBs, which help anchor the placenta to the uterus. In the distal parts of such anchoring villi, the CTBs differentiate into a migratory phenotype known as EVT. During CTB differentiation toward EVT differentiation, the Integrin alpha-6/beta-4 (ITGA6:ITGB4) is changed to the villous integrin alpha-5/beta-1 (ITGA5:ITGB1) in cell columns and α1β1 in the uterine wall. The differentiated EVTs are characterized by the absence of epidermal growth factor receptor (EGFR) expression. On the other hand, there was an increase in the positivity of HLA-G expression on their surface [35]. Integrin expression is reflected in altered cell adhesion properties. Previously, the expression of tumor protein (TP63) in the human placenta has been described and shown to be expressed only in proliferative CTBs and completely eliminated from both STBs and EVTs [35,53]. The α isoform of the N-terminally truncated p63 (ΔNp63α) has been shown to be involved in maintaining the stem cell state in stratified epithelia.

### 2.5. The Role of the TGFβ Superfamily in Placentation 

The transforming growth factor-beta (TGF-β) contains a family of dimeric polypeptides that affect the differentiation and proliferation of several cell types. TGF-β acts as a proangiogenic factor. Angiogenesis is strictly regulated between pro-and anti-angiogenic factors. The change in the ratio of acting factors can lead to pathological processes. TGF-belongs to the regulatory cytokine superfamily, which has pleiotropic functions in many cell types and is involved in many physiological and pathological processes in the vertebrates [54]. Smads are intracellular nuclear effectors of TGF-β family members. TGF-β has a biological effect through binding to cell surface receptors. The Smad family can be divided into three distinct subfamilies: type I receptor-regulated (R-Smads), type II receptor common-partner Smads (Co-Smads) and type III receptor inhibitory (I-Smads). Smad complexes can bind to DNA through other DNA binding proteins and regulate the transcription of target genes [55]. TGF-β regulates a variety of developmental and physiological processes, including placental development. TGF-β plays an important role in the precise regulation of trophoblast proliferation, differentiation and deciduous invasion. With an adequate balance in the functions of all its members, the cytokine improves the course of placentation, and an imbalance can cause aberrant placentation and more related pregnancy complications. At present, the complete function of members of the TGF-β superfamily in the regulation of placentation has not been elucidated. Authors admit the possibility of their dysregulated expression in case of placental disorders and pregnancy complications. TGF-β signals and their individual subunits. Reports on the role of TGF-β in human TB invasion are contradictory. TGF-β may promote trophoblast invasion into decidua by increasing the expression of matrix metalloproteinase-2 (MMP2) and MMP9. TGF-β, on the other hand, may inhibit trophoblast invasion into the decidua by decreasing MMP2 and MMP9 expression, decreasing fibronectin deposition, primarily by increasing tissue TIMP1 and cyclooxygenase-2 (COX-2) expression [50]. Authors observed increased expression and activity of COX-2 in the trophoblast cells of women compared to healthy pregnant women with preeclampsia. It has been shown that COX-2 and prostaglandin E2 (PGE2), which is mediated by COX-2 in conjunction with TGF-β1, can inhibit the invasion of human trophoblast cells. It is not clear whether TGF-β1 induces COX-2 in human trophoblast cells or whether COX-2 mediates the suppressive effects of TGF-β1 on trophoblast invasion. COX-2 expression is usually low but can affect many factors, such as cytokines, hormones and growth factors. COX-2 is basically expressed in small amounts, including in the placenta and brain [56,57]. Authors describe the treatment of acetylsalicylic acid in patients at high risk of preeclampsia. Acetylsalicylic acid is known to inhibit COX-2, which is responsible for prostaglandin synthesis [58].

### 2.6. Immunomodulation in Early Placentation 

Pregnancy is physiologically controlled by the immune system to ensure the development of semialogenic fetus. If these processes are not properly adapted, miscarriage or preeclampsia are likely to occur [59]. The initial step of spiral arterial remodeling is likely to be regulated by immune cells that are localized in vessels before EVT involvement [57]. Maternal decidua in normal pregnancy contains several types of immune cells, such as macrophages, NK cells and regulatory T cells (Treg). The majority of these cells are decidual NK cells, and a minority are macrophages and dendritic cells [60]. Factors released from the placenta affect the balance of maternal cytokines. Syncytiotrophoblast microvesicles (STBMs), which are secreted into the maternal circulation, affect monocytes and B cells and induce the release of tumor necrosis factor-alpha (TNF-α) and interleukins IL-1α, IL-1β, IL-6, IL-8. In the last trimester, human peripheral blood mononuclear cells (PBMC) produce, in response to STBMs, more TNF-α and IL-6 in pregnant women than PBMC in non-pregnant women. It is hypothesized that maternal immune cells are primed by pregnancy, presumably through their interaction with STBMs [59]. Decimal NK cells indirectly regulate trophoblast invasion in vitro and in vivo by producing IL-8 and interferon-inducible protein-10 chemokines (CXCL10). dNK cells are potent secretors of a number of angiogenic factors with the ability to induce vascular growth in the decidua. Thus, the regulation of trophoblast invasion by dNK cells makes it possible to indirectly regulate vascular remodeling [61].

## 3. In Vitro Placental Models

### 3.1. Primoculture Trophoblasts Monolayer Cells

The placenta’s in vitro model allows the modeling of various cellular and metabolic processes inside the human body. However, the ability to detect early processes in placentation and the mechanisms by which human TE function is specific, is currently poorly understood. Several authors cultured trophoblast cells in the past. Traditionally, cell cultures have been performed using two-dimensional (2D) systems where cells grow in a monolayer. Researchers isolated trophoblast cells from the scattered placental tissue that was not contaminated with blood elements, macrophages and mesenchymal cells using Percoll gradients [39]. The populations of mononuclear cells of placental tissue were obtained in the samples, which showed structural properties, as well as some biochemical properties of trophoblasts. Cells harvested from the middle gradient band appeared round, with the majority having a diameter of about 10–20 μm [39]. Scientists still use the isolation of primary cells from placental tissue. From the terminal placental tissue following the enzymatic digestion, Percoll centrifugation makes it possible to isolate CTBs, which, following in vitro culturing, form multinucleated structures characterized by upregulated markers of STB identity, such as hCG [62]. The growth of placental cells from the explants is provided as another possibility. Explant cultures have grown from smaller pieces of placental tissue, several millimeters in size. Individual tissue sections were immersed in the culture medium and the cells proliferated from the explanted tissue. The growth of placental cells via explants is facilitated in hypoxic conditions, where cultivation in closed flasks promotes rapid growth. Under the above-mentioned conditions, cytotrophoblast proliferation increases. Throughout this procedure, it was possible to prepare cultures of trophoblastic cells that replicate, sometimes in restricted mode, for two to three cycles, excluding chorionic gonadotropins. The best results were achieved when the original placental explants were obtained from the CTBs of the first-trimester placental column, in contrast to the wounds where the villi pieces are located [63].

### 3.2. Human Cancer Cell Lines

Human cell lines are used in experiments for extended periods. The human choriocarcinoma cell line JAGs are known cell lines that have endocrine production, secreting gonadotropin, human chorionic somatomammotropin, progesterone, estrogen and estradiol. The JEG-3 line is partially similar in functionality to the placental tissue for trophoblast invasion in vitro [64]. Human choriocarcinoma cell lines, BeWo cells, began to be used in the 1980s as an in vitro model for the placenta. The b30 subclone BeWo cells can grow on a membrane system to form confluent cell layers. These layers allow testing of metabolism in the placenta [65]. HTR-8/SVneo was developed in the first trimester using an EVTs infected with simian virus T antigen (SV40). HTR-8/SVneo cell lines are widely used to study trophoblast functions, including cell fusion, migration and invasion. The purity of each cell line is therefore crucial so that it can be used as a model of recapitulating trophoblast cells [66]. The JEG-3 human choriocarcinoma cell line is grown in the form of a single cell suspension cultured without serum. In addition, JEG-3 and JAR, cancer cell lines with an abnormal number of chromosomes have, in some cases, distinctly different transcriptomic profile compared to EVT [2]. In an in vitro invasion assay, authors demonstrated the significant invasive capacity of the JEG-3 cell line, which was compared to the JAR cell line. It was further found that the expression of heparanase mRNA protein in human choriocarcinoma cells JEG-3 and JAR was clearly higher than in normal chorion [67]. The JEG-3 cell line expresses endogenous HLA-G, which is useful as a positive control for the EVT layer [68,69,70]. JEG-3, BeWo, JAR, HTR-8/Svneo and Swan-71 cell lines are the most commonly used for experimental testing of trophoblast migration [2].

### 3.3. Trophoblast Stem Cells of the Blastocyst

Human stem cells can currently be isolated by several methods. Mammalian blastocyst consists of two types of cell layers, the outer TE surrounded by pluripotent cells, forming the inner cell mass (ICM). Blastocysts give rise to three stem cell entities—the pluripotent ESCs, which are derived from ICM developed from the epiblast, and two types of extraembryonic stem cells, primitive eXtraembryonic ENdoderm-derived (XEN) cells and TSCs derived from extraembryonic ectoderm. Human embryonic stem (hESCs) cells are usually derived from the ICM of blastocysts. The derivation of hESCs is therefore considered ethically controversial, and the embryo is destroyed after isolation. Multipotent polar TE cells later develop into the embryonic part of the placenta. Stem cells of all three lines, ESCs, XEN and TSCs, self-renew and retain their fate-specific development potential in vitro [71]. TSCs were derived from TE blastocysts or extraembryonic ectoderm following the implantation [72]. Several authors describe successful cultivation of the TSC line, which was derived from human embryonic blastomeres and chorionic mesenchymal cells and cultured with 10% fetal bovine serum and fibroblast growth factor 2 (FGF2) [23,73,74]. Recently described TSCs from the blastocyst are bipotential. These cells are capable of differentiating into vCTBs and STBs and are suitable for the formation of 3D epithelial organoids, closely resembling the structure and physiology of the original organ (Figure 4).

### 3.4. Induced Stem Cell Engineering Cell Fate

In mammals, cell fate segregation takes place shortly after fertilization, when the outer morula cells specialize in the future trophoblasts. Embryonal cell fate is controlled by several stimuli, including cell polarity and position, cell–cell signaling and differential expression of TFs. Unlike the in vivo environment, the in vitro conditions are different. Embryonic cell does not spontaneously differentiate into extraembryonic cell types. In an in vitro environment, it is possible to target the fate of cells with specific TFs. Induced PSCs can be generated from somatic cells or adult fibroblasts after forced expression of pluripotent TFs. Similarly, direct reprogramming to multipotent trophoblastic stem cells can be induced. It has been shown that hTSCs can be generated by somatic cells by two methods of reprograming—either adult fibroblasts with the TFs SOX2, Octamer-binding transcription factor 4 (Oct4), Krüppel-like factor 4 (KLF4) or the transformation of PSCs. Human-induced PSCs, which have been generated by reprograming somatic cells, are capable of differentiating into a wide range of body tissue types [72]. It has been shown that a reduced Oct4 expression induces trophoblastic morphology. In the overexpression of the Oct4 antagonist, TF CDX2 is able to induce a trophoblast, its morphology and upregulation of trophoblast markers. Eomes, a factor behind CDX2, is overexpressed and affected by differentiation toward TE/TSC, making both CDX2 and Eomes strong candidates for key TE regulators. Decreased expression of Oct4 in ESCs resulted in the loss of pluripotency and the formation of a monolayer by trophoblast-like cells. CDX2 and Eomes have a key effect on the regulation of differentiation into the trophoblast line. TEAD4 acts on CDX2 during preimplantation, leading to the initiation of TE formation [71] (Table 2). Furthermore, TFs were identified as inevitable for the successful induction of TSCs, GATA3, Eomes and the transcription factor TFAP2C. Eomes and TFAP2C bind to the TEAD4, GATA3 and E74-like ETS transcription factor 5 (ELF5) genes and positively regulate the expression of ELF5, a protective gene in trophoblastic line differentiation in the early blastocyst. The triple casts of ELF5, Eomes and TFAP2C were enriched for TSC proliferation and potency of TSCs. GATA3 binds to expression and upregulates CDX2, which also increases self-expression as ELF5 and Eomes. GATA3 also induces Eomes expression independently of CDX2 [75]. When it comes to comparing the molecular differences in humans and rodents, known molecular differences include the timing of the CDX2 expression, a key TE TF that is detected in humans only after blastocyst formation. GATA3 expression is more pronounced in human TE, presumably to compensate for the late CDX2 expression. Differences in expression patterns also exist between human and mouse blastocysts. For example, Oct4 is ubiquitously expressed in all human blastocyst cells on days 5–7, while in mice, it is restricted to the ICM [76]. TSCs are characterized by the morphological, transcriptional and functional characteristics of the human cytotrophoblast in vitro. TSC self-renewal could be maintained on a layer of inactive mouse embryonic fibroblast feeder cells that are provided with factors related to the NODAL pathway and supplemented with FGF4, heparin and fetal bovine serum. The induction of human trophoblast progenitors is an important factor. TSCs could be maintained under these conditions for several passages [77]. A technique has recently been described for inducing human-induced trophoblast progenitor (iTP) cells by directly reprogramming fibroblasts with murine trophoblast line-specific transcription factors consisting of CDX2, Eomes and ELF5. Human iTP cells were given epithelial morphology and could be maintained in vitro for more than 2 months [44]. The authors proposed the recommended criteria for the TSC phenotype of GATA3, KRT7 and TFAPC2 markers that lack the expression of HLA class I, ELF5 promoter hypomethylation and C19MC expression. In addition, DNA methylation deficient progenitor cells can efficiently differentiate into trophoblast-type cells [2,62,78].

### 3.5. Trophoblast Organoids and Spheroids as Placental Model

#### 3.5.1. Placental Tissue Culture

More sophisticated in vitro cultivation methods are currently available. In 3D cultures, cells can proliferate and differentiate into a networked environment of biological or synthetic materials that simulate natural ECM. Human TSCs were successfully derived from the blastocyst in the first trimester VCT. Authors analyzed primary transcriptomes of human trophoblast cells in order to deduce how CTBs are maintained in their undifferentiated form in an in vivo condition. Culture conditions and derived human TSCs from CTBs and blastocysts were optimized. Line markers, such as alpha 6 integrin subunit (ITGA6) and TP63, were confirmed on cytotrophoblasts, including ITGA5, HLA-G extravillous trophoblasts, placental lactogen (CSH1) and chorionic gonadotropin (CGB) syncytiotrophoblasts. A 3D model of human TS cells from the blastocyst using a suitable Matrigel was created, and growth of EVT cells from the placental explants was induced. TS cells were cultured for a long time, and the cells expressed the markers KRT7, TP63, GATA3, TEAD4. The markers CGB and SDC1 were highly expressed on syncytiotrophoblasts. On the contrary, HLA-G and vimentin were weakly expressed [62]. Placental organoids mimicking the human trophoblast include cells of extraembryonic origin. Primary culture can be used to create isolated organoids. It was shown that a suspension of trophoblast-enriched cells was obtained in the first trimester by enzymatic cleavage of villi from the placental tissue. Placental villous stromal cells (PVSCs) were isolated by digestion of the tissue remaining, following the initial trypsin and collagenase digestion. The formation of small organoid clusters that formed after 7–10 days was observed. Trophoblast organoids were passaged and placed on differentiation plates. Then, they were maintained in the trophoblast organoid differentiation medium [18]. TSCs with the ability to differentiate into STBs and EVTs are ideal for generating organoids. An in vitro generation of human cytotrophoblastic organoid cultures (CTB-ORGs) from a purified first trimester capable of self-renewal and expansion under defined culture has been shown. Purified first-trimester CTBs were applied to human trophoblast organoids from placental tissues and subsequently integrated into a Matrigel matrix containing a mixture of growth factors and signaling inhibitors [23,78]. Matrigel is widely used to induce the growth of EVT cells from the perinatal tissue [23,79]. Trophoblast culture medium (TOM) composed of EGF, FGF2, CHIR99021 (WNT activator), A83-01 (TGFβ/SMAD inhibitor) and R-spondin-1 was used to culture EVT cells [80]. Early placental proliferative 3D structures expressed markers of the human CTB strain in the epithelial cell layers and spontaneously underwent cell fusion toward the center [23]. Self-renewing trophoblast organoids grow as complex structures that accurately recapitulate the structure of the placental villi in vivo, where vCTBs expresses the markers EpCAM (epithelial cell adhesion molecule) and cadherin-1 (CDH1) [23]. Organoids have been shown to express CDX2, TP63, TFAP2C, TFAP2A, TEAD4 and GATA3 in the outer layers of CTB-ORG [23,81]. Lee’s authors report a phenotype characteristic of human cytotrophoblastic organoid cultures (CTB-ORG) and primary first-trimester trophoblasts, HLA class I profile, ELF5 methylation and microRNA (miRNA) expression from the miRNA cluster chromosome 19 (C19MC) [78]. Integrin P1 proliferative markers are characteristic of human TP niches in the first trimester of the placenta [27]. The differentiation process of early extraembryonic tissues is important in monitoring the processes of placentation in vitro. Human PSCs are used in the study of trophoblast differentiation [82]. In vitro techniques mimic in vivo ones, where TSC also differentiates into EVT, a process crucial for proper placentation. The advantage of the 3D model is the arrangement into complex structures generating both STBs and EVT. The formation of the simulated natural structure of the placental villi can be closely monitored histologically. The lacunae present in the syncytial regions are similar to those found in vivo [23,80]. Authors’ data now provide functional evidence that naive hPSCs indeed have enhanced potential to access trophoblast fates during both spontaneous differentiation assays and upon treatment with recently devised conditions for hTSC isolation [62,83]. Other authors describe a sheep placenta model in a series of studies analyzing the dynamic interrelationships between the trophoblast and the uterine epithelium. Trophoblastic epithelium consists mostly of typical trophoblast cells and minimal binucleate trophoblast cells derived from them. Binucleate cells synthesize and accumulate placental lactogen [84,85]. 

#### 3.5.2. Organoids Mimic EVT 

Trophoblast migration through the maternal decidua and remodeling spiral arteries affects the extracellular environment. Matrix-integrin binding induces a signal transduction cascade. Such process plays a key role in controlling the cellular behavior and expression of MMP2 and MMP9, which facilitate collagen skeletal degradation. The degradation of ECM EVTs can be reproduced in vitro, where the cells degrade the Matrigel layer on which they are cultured. Authors used an EVT-like cell line HTR8/SVneo, which in combination with Matrigel induces differentiation to the EVT phenotype [86]. The phenotype with EVT differentiation potential includes the following markers: cytokeratin 7 (KRT7), CD9 trophoblast stem cell lines EVT proteases MMP2, MMP9, ECM degradation inhibitors TGFB1, adhesion molecules TGFB2, ITG extracellular EGA and extracellular ITHk, cadherin CDH5 [86].

#### 3.5.3. Spheroids of Placenta-Derived Mesenchymal Stem Cells 

Placenta-derived mesenchymal stem cells (MSCs) are used for regenerative medicine thanks to their multilineage potential and regenerative properties. MSCs can be isolated from extraembryonic tissues, umbilical cord, placental tissue, amniotic and chorionic membranes, where they are involved in maintaining stem cell and tissue niches [87,88,89,90,91]. PMSCs have a characteristic viability and proliferation [92]. MSCs preparation as spheroids is a method used for optimizing the improvement in the efficacy of MSC-based therapeutics. Culturing MSCs in the form of three-dimensional spheroids is a simple and reproducible method and has several advantages over the 2D monolayer culture [93]. Human amniotic mesenchymal stem cells (hAMSCs) were cultured in a serum-free DMEM medium. Mesenchymal phenotype of CD90+, CD73+, CD13+, CD45—and HLA-DR—was confirmed in the tested cells [94]. The tested hAMSC spheroids had increased expression of angiogenic and growth factors HGF, PDGF, VEGF, FGF1, EGF, as well as immunosuppressive factors IL6, TGF-β, COX2. It has been detected that culturing MSCs in the form of spheroids was suitable for maintaining multipotency and paracrine production of hAMSCs [94]. MSCs have a therapeutic effect either due to their requirement for replacement cells or their immunomodulatory and paracrine activities, which promote tissue regeneration. MSCs appear to be the most effective treatment option for patients suffering from musculoskeletal diseases [95,96] (Table 3).

#### 3.5.4. Characteristics Phenotype of Trophoblast Organoids

Specific trophoblast markers make it possible to recognize the degree of cell development and cell differentiation. Phenotyping of organoid markers of trophoblasts is very important to determine their defined phenotype. The authors Turco et al. recommend the characterization of the following markers: HLA-ABC, HLA-G, ITGA2 [18]. Transcription factor GATA3 affects the embryonic development of various tissues and acts in inflammatory and humoral immune responses. KRT7 stimulates DNA synthesis in cells. KRT7 is expressed by EVT, specifically STBs and vCTBs, which express HLA class I. HLA-G positivity is characteristic of EVT cells, thus it is an accepted specific marker detected for EVT identification [68]. Trophoblast cells from animal placenta express cytokeratins II (KRTII). Trophoblast cells are distinguishable from syncytia cells, which had a very low expression on KRTII [85]. Interestingly, the inhibitors of placental differentiation have been described. A study investigated the effect of transforming growth factor β1 (TGFβ1) on cytotrophoblast differentiation. It has been proven that TGFβ1 acts as a major inhibitor of trophoblast differentiation and concomitant peptide hormone secretion [97]. Invasive trophoblasts have a capability to provide chemotactic signals to uterine leukocytes and affect decidual angiogenesis and apoptosis by secreting hCG [29]. CDX2 is essential for trophoblastic development, vasculogenesis in the mesoderm of the yolk sac, allantoic growth and chorioallantoic fusion [98]. CDX2 is expressed in a mouse model for 3.5 days post-coitum in the trophectoderm region. Along with CDX1 and CDX4, CDX2 is one of three caudal-related genes in the human genome. Genes are also present in most vertebrates’ genomes. CDX2 is thought to play an important role in the pathways controlling the embryonic axial elongation and anterior-posterior patterning [98]. TP63 is a member of the p53 family of transcription factors, and TP63 is involved in skin development and regulation of ASCs. TP63 encodes a transcript isoform that plays a role in ovarian germ cell survival [99]. It has been found that human stem cells induce trophoblasts by induction of BMP 4, leading to the formation of cells with the p63+/KRT7+ phenotype, which is a stem CTB. It has also been observed that p63 expression in CTB cultures had an inhibitory effect on the secretion of hCG [35]. The presence of transcription factors involved in the regulation of trophoblast development has been noted. TFAP2A, TFAP2C and GATA3 in CTB organoids were also expressed in STB and CTB placentas in the first trimester. The observed mRNAs were also detectable in CTB-ORG [23,48,49]. 

## 4. Conclusions

Organoids are currently a very promising culture technology with more advantageous properties compared to traditional cancer cell lines and primary cultures [100]. An organoid system allows researchers to intensively study the processes that control embryonic development, tissue specification and tissue homeostasis, as well as the onset and manifestation of disease. As organoids are formed from ESC, iPSC and fetal tissues, they faithfully preserve the properties of their original developmental stage. As a result, detailed images of embryonic development in the dish can be obtained. Cell differentiation is systematically invoked. Organoids provide an important insight into the development of stem cells and their niches. At the same time, they allow us to monitor their differentiation into mature functional lines. Organoid cultures can be obtained from human ESCs or adult progenitor cells [5]. The main advantages of the mentioned organoids are as follows—they can be maintained in culture for a long period of time with a continuous passage, and they can also be stored by cryopreservation [80].

## Figures and Tables

**Figure 1 biomedicines-10-00904-f001:**
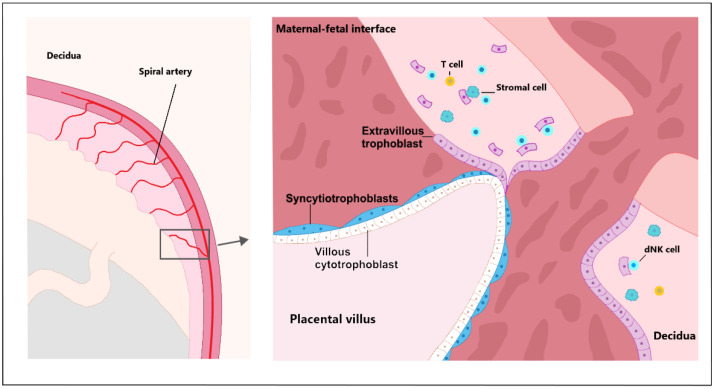
Trophoblast stem cells differentiate into more specialized trophoblast populations at an early stage of development. TSCs are further divided into mononuclear cytotrophoblasts and multinucleated primitive syncytium. Cytotrophoblasts have the ability to proliferate, differentiate and fuse with the syncytiotrophoblast, thereby promoting syncytial growth during ontogenesis. Cells of the cytotrophoblast divide and migrate externally. Through cell fusion, the cytotrophoblast forms a multinucleated syncytiotrophoblast. Thus, cytotrophoblasts are the basis of the syncytiotrophoblast and reside on the basement membrane that separates them from the villous stroma. Extravillous trophoblastic cells line the mother’s blood vessels and intersect with maternal cells in the decidua, as well as several types of maternal immune cells, including T cells, decidual natural killer cells (dNK) and stromal cells that provide structural support for the decidua.

**Figure 2 biomedicines-10-00904-f002:**
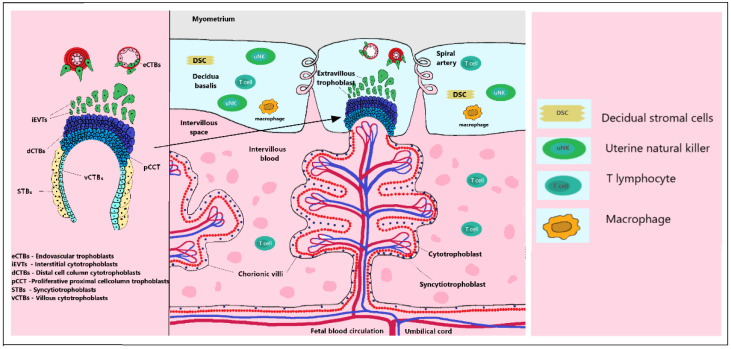
Maternal spiral arteries deliver nutrients to the placenta. Structure of a placental anchoring villus and its different trophoblast subtypes. Precursors that reside in the villous cytotrophoblast (vCTB) layer either differentiate into multinuclear syncytiotrophoblasts (STBs) when surrounded by maternal blood or give rise to proliferative proximal cell column trophoblasts (pCCTs) upon attachment of villi to the maternal decidua. Following the differentiation into distal cell column trophoblasts (dCCTs), extravillous trophoblasts (EVTs) develop, breaking through the overlying STB layer. Differentiation of EVT subtype progenitors residing on the basement membrane of CCTs leads to interstitial cytotrophoblasts (iCTB) and endovascular cytotrophoblasts (eCTB) relocated to maternal helical arteries. iCTBs in the decidual stroma colonize blood vessels from the outside and communicate with uterine cells of various types, such as decimal stromal cell macrophages and uterine natural killer (NK) cells.

**Figure 3 biomedicines-10-00904-f003:**
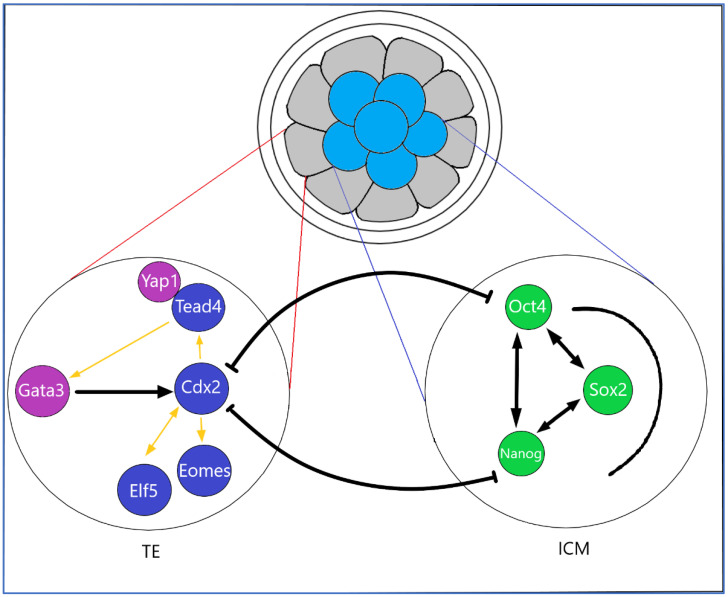
Pregnancy is characterized by elevated levels of specific cytokines at the fetal–maternal interface. Blastocysts contain pluripotent ESC stem cells that are derived from the inner cell mass (ICM). Complicated differentiation process of the trophoblast precursors in the trophectoderm (TE) of early ontogenesis is modulated by transcription factors, which act as transcriptional promoters, leading to the formation of the fetal part of the placenta. Placenta-specific transcription factors are involved in further trophoblast development via transcription of placenta-specific genes. At the beginning of pregnancy, there is a transcriptional enhancer factor 3 (TEAP4) binding Yes-associated protein (YAP) coactivator, which plays a key role in the proliferation and expression of trophoblast stem cells (TSCs) villous trophoblastic epithelium progenitors. Caudal-type homeobox 2 (CDX2) plays a crucial role in the development of fetus and its perinatal tissues. In the overexpression of the octamer-binding transcription factor 4 (Oct4) antagonist, transcription factor CDX2 is able to induce a trophoblast, its morphology and upregulation of trophoblast markers. Eomesodermin (Eomes), a factor behind CDX2, is overexpressed and conditions differentiation toward TE/TSC, making both CDX2 and Eomes strong candidates for key TE regulators. Decreased expression of OCT4 in ESCs results in the loss of pluripotency and the formation of a monolayer by trophoblast-like cells. CDX2 and Eomes have a key effect on the regulation of differentiation into the trophoblast line. Tead4 acts on CDX2 during preimplantation, leading to the initiation of TE formation.

**Figure 4 biomedicines-10-00904-f004:**
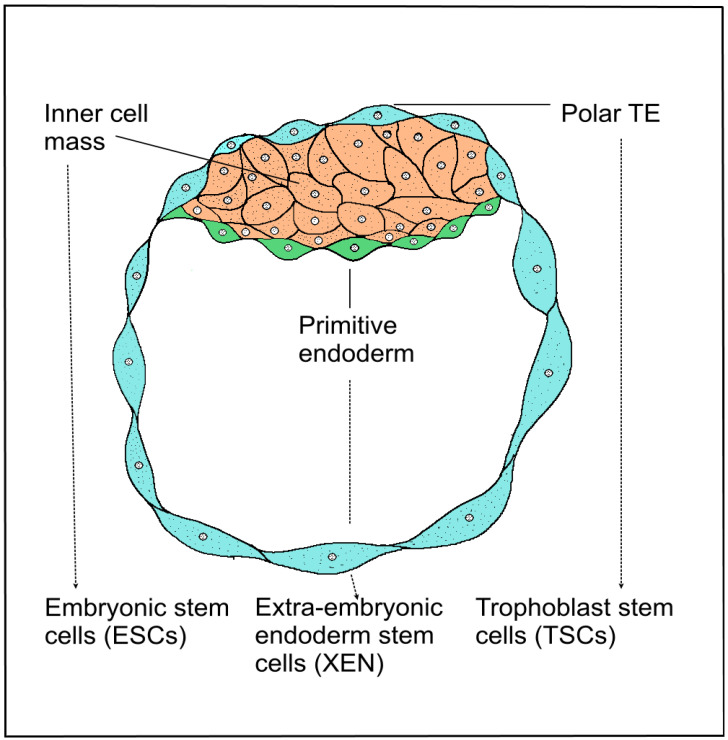
Schematic development of a blastocyst. Mammalian blastocyst consists of two types of cell layers, the outer trophectoderm surrounded by pluripotent cells, forming the inner cell mass (ICM). Blastocysts give rise to three stem cell entities—the pluripotent embryonic stem cells (ESCs), which are derived from ICM developed from the epiblast, and two types of extraembryonic stem cells, primitive eXtraembryonic ENdoderm-derived (XEN) cells and trophoblast stem cells (TSCs) derived from extraembryonic ectoderm. Solid lines—description of blastocyst cell layers. Dotted lines—three types of blastocyst-derived stem cells.

**Table 1 biomedicines-10-00904-t001:** Abbreviations often used in the text.

Abbreviation/Glossary	Acronym
Adult stem cells	ASCs
Cell column trophoblasts	CCTs
Cytotrophoblast	CTBs
Decidual natural killer	dNK
Distal cell column cytotrophoblast	dCTBs
Embryonic stem cells	ESCs
Endovascular cytotrophoblasts	eCTBs
Extravillous trophoblasts	EVTs
Human embryonic stem	hESCs
Inner cell mass	ICM
Interstitial cytotrophoblasts	iEVTs
Mesenchymal stem cells	MSCs
Placental villous stromal cells	PVSCs
Pluripotent stem cells	PSCs
Primary villi	PV
Primitive syncytium	PS
Proliferative proximal cell column trophoblasts	pCCTs
Syncytiotrophoblasts	STBs
Syncytiotrophoblast microvesicles	STBMs
Trophectoderm	TE
Trophoblast progenitor cells	TPCs
Trophoblast stem cells	TSCs
Villous cytotrophoblasts	vCTBs

**Table 2 biomedicines-10-00904-t002:** Specific markers identified in the early stage of implantation and different trophoblast cell subtypes.

Cell Type	Phenotype	References
Morula cells (Mrl)	Elf5, EOMES GATA6, BMP4, SOX17	[6]
Inner Cell Mass cells (ICM)	Nanog, Oct4, SOX2	[71]
Trophectoderm cells (TE)	FGF4	[46]
Cytotrophoblasts (CTBs)	Tpbpa, CDX2, GATA3, TFAP2C, TEAD4, E-cadherin, CK7	[23,41]
Villous Cytotropboblasts (vCTBs)	GATA3, CDX2, TP63, TEAD4, K167, ITGB1, E-cadherin	[18,23,34]
Syncytiotrophoblast (STBs)	GATA3, TFAP2A, TFAP2C, hCG, EGFR, hCG	[18,23]
Extravillous Trophoblasts (EVTs)	NOTCH1, CEA adhesion molecule 1, EGFR	[34]
Cell Column Trophoblasts (CCTs)	NOTCH1, CK7, E-cadherin, VE-cadherin	[1,34]
Distal Cell Column Trophoblasts (dCCTs)	NOTCH1, NOTCH2, HLA-G	[34,49]
Endovascular Cytotrophoblasts (eCTBs)	CK7, VE-cadherin, PECAM	[1]
Intersticial Cytotrophoblasts (iCTBs)	ITGA1, MMP 12, CK7, HLA-G, ITGBA1B1	[1,34]

Abbreviations list. Caudal-type homeobox 2 (CDX2), Cytokeratin 7 (CK7), GATA binding protein (GATA), Integrin beta 1 (ITGB1), human chorionic gonadotropin (hCG), TEA-domain transcription factor 4 (TEAD4), tumor protein p63 (TP63), matrix metalloproteinase (MMP), E74-like ETS transcription factor 5 (ELF5), Eomesodermin/T-box brain protein 2 (Eomes), SRY-related high-mobility-group box (Sox), octamer-binding transcription factor 4 (Oct4), transcription factor AP-2gamma (TFAP2C), transcription factor AP-2alpha (TFAP2A), epidermal growth factor receptor (EGFR), neurogenic locus notch homolog protein (NOTCH).

**Table 3 biomedicines-10-00904-t003:** Advantages and disadvantages of different cell culture techniques and cell placenta populations listed in chronological order.

Type	Cell Type	Advantages	Disadvantages
Primo Culture Monolayer Trophoblasts	Trophoblast cells	Simple isolation	Differences in cell morphology in vivo and in vitro data
		High viability during cultivation	Lost characteristics during culture periods
		Large number of cells by subculturing	Time-limited cell growth data
Human Cancer Cell Lines	Choriocarcinoma cell lines	Unlimited cell growth	Chromosomal aberrations Abnormal number of chromosomes
		Large numbers of cells by subculturing	
	JAGs	HLA-G expression	
	JEG-3	Trophoblast invasion in vitro	
	BeWo	Testing of metabolism	
	HTR-8/Svneo	Cell fusion, migration and invasion	
Trophoblast Stem Cells of Mouse Blastocyst	XEN	Self-renewing and retaining their fate-specific development potential in vitro vCTBs and STBs differentiating	Embryo is destroyed after isolation
	ESCs		
	TSCs		
Induced iPSCs	Induced human trophoblast stem cells (iTPs)	Differentiating into a wide range of body tissue types Suitable for the study of disease mechanisms	Efficiency of reprogramming is generally low
Spheroids of Placenta Cells	Placenta-derived MSCs Amnion MSCs Chorion MSCs Wharton-Jelly-derived MSCs	Easy-to-use protocol Co-culture ability High reproducibility	Simplified architecture
Placental Organoids	Human CTB-ORGs Human TSCs	Organoid mimic tissue architecture relatively easy to grow Suitable for the study of the human placenta An organoid ideal for development and function mimics EVT	Heterogeneity of organoids, problem with the uniformity of cell population Lack vasculature

Abbreviations list. Extra embryonic endoderm-derived (XEN), Human embryonic stem (ESCs), Trophoblast stem cells (TSCs), Induced pluripotent stem cells (iPSCs), Mesenchymal stem cells (MSCs), Villous cytotrophoblasts (vCTBs), Syncytiotrophoblasts (STBs), Human cytotrophoblastic organoid cultures (CTB-ORGs), Extravillous trophoblasts (EVT).

## Data Availability

Not applicable.

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
