# Peer review of "In Vitro Model of Human Trophoblast in Early Placentation"

_biomedicines, 2022, doi:10.3390/biomedicines10040904_

Round 1
Reviewer 1 Report
The review entitled “In vitro model of human trophoblast in early placentation” by Bačenková and colleague aims to highlight the knowledge on the development of human placenta and different in vitro model system for the study of placental physiology. Overall, the manuscript is well-written and easy to follow. I have no major comments, however, I have some minor suggestions for the authors:
Please make a table with details of stem, syncytiotrophoblast and extavillous trophoblast cell markers.
Section 2.2 Early hypoxic placental environment: Please also include the importance of HIF1B and HIF2A in placental development.
Line 327, Cited reference 60, Okae et al also used 3D culture for syncytiotrophoblast. Please include the some details about the Okae et al derived human trophoblast stem cells in this section.
Line 559 and 564-565 looks similar.
Author Response
Dear Sir,
thank you very much for your comments to topic. Following your instructions, entire text was checked thoughly and corrected accordingly. Attached please find a corrected version of the review.
Looking forward to your reply.
Kind regards, Darina Bačenkova
Answer 1.:
In the appendix attached please find a table with the details syncytiotrophoblast and extravillous trophoblast cell marker.
Answer 2.: Text related to the importance of HIF1B and HIF2A has been added. Below citation has also been added.
Answer 3: A text has been added to the Placental tissue section on the 3D culture for the syncytiotrophoblast, citing the text of Okae et al.
Answer 4: Sentence with similar content was deleted.

Reviewer 2 Report
The review seems interesting, but I have a number of comments.
It is necessary to accurately describe in the Introduction what the purpose and objectives of this review are. In addition, its novelty is unclear. It is necessary to describe whether reviews on related topics have been published and, if so, to refer to them and justify the novelty of your research. It is also necessary to justify the structure of the review.
Information about the models used in vitro should be additionally presented in the form of a table for better understanding. Also, the authors need not only to list the available in vitro models, but also to analyze their limitations. It is necessary to analyze which models are suitable for which research purposes. Without such an analysis, the presented review has no serious value.
I also saw a few typos and I think that the authors need to carefully check the entire text.
Author Response
Dear Sir,
thank you very much for your comments to the topic. Following your instructions, entire text was checked thoroughly and corrected accordingly. Attached please find a corrected version of the review.
Answere 1: Text concerning the relevance of the arcicle´s content has been added. Also some autors´ notes on the subject have been included.
Answere 2: Table with an overview of in vitro culture types of placental cells has been added. Information on the advantages and disadvantages of individual in vitro placental cell models was added.
Answere 3: The entire text has been thoughly rechecked.

Round 2
Reviewer 2 Report
The authors improved their manuscript in accordance with my recommendations. I believe that the revised manuscript may be published.